# The Effects of Counter-Ions on Peptide Structure, Activity, and Applications

**DOI:** 10.3390/biom15111567

**Published:** 2025-11-07

**Authors:** Ying Liu, Yi Huang, Lan Yang, Yu Gao, Zheng Jia, Tingting Liu, Baoling Su, Chuyuan Wang, Lili Jin, Dianbao Zhang

**Affiliations:** 1School of Life Sciences, Liaoning University, Shenyang 110036, China; 4022510309@smail.lnu.edu.cn; 2Department of Stem Cells and Regenerative Medicine, Key Laboratory of Cell Biology, National Health Commission of China, and Key Laboratory of Medical Cell Biology, Ministry of Education of China, China Medical University, Shenyang 110122, China; huangyi18004010579@163.com (Y.H.); yanglan1@cmu.edu.cn (L.Y.); gaoyu@cmu.edu.cn (Y.G.); 18759616564@163.com (Z.J.); 3Department of Pharmacy, Liaoning Agricultural Technical College, Yingkou 115009, China; liutingting19881123@hotmail.com; 4College of Life Science and Bioengineering, Shenyang University, Shenyang 110044, China; sblwjg@163.com; 5Department of Endocrinology and Metabolism, The First Affiliated Hospital of China Medical University, Shenyang 110001, China; chuyuan0215@163.com

**Keywords:** counter-ions, peptide, biological activities, peptide delivery, chromatography

## Abstract

Peptide drug development has emerged as a prominent area in pharmaceutical research due to its high specificity and therapeutic potential. However, their biological activity, stability, and bioavailability are significantly influenced by interactions with counter-ions, which electrostatically bind to charged residues on peptide surfaces. This review systematically examines the multifaceted roles of counter-ions in modulating peptide structure and function. Counter-ions are classified into organic/inorganic and anionic/cationic categories, with their selection critically impacting peptide solubility, conformational stability, and activity. Inorganic counter-ions could enhance structural integrity, while organic counter-ions could mitigate toxicity risks. Notably, counter-ions can induce secondary structural transitions, directly affecting biological efficacy. Furthermore, counter-ions play pivotal roles in drug delivery systems, including nanoemulsions, self-emulsifying formulations, and lipid-based nanoparticles, where hydrophobic ion pairing improves encapsulation efficiency and oral bioavailability. In chromatography, ion-pairing reagents optimize peptide separation but may compromise mass spectrometry compatibility. Emerging analytical techniques, such as capillary electrophoresis and liquid chromatography–tandem mass spectrometry (LC-MS/MS), enhance counter-ion detection precision, addressing challenges in pharmaceutical quality control. Despite advancements, gaps remain in understanding ion-specific binding mechanisms and long-term safety profiles. This review underscores the necessity of tailoring counter-ion selection to balance efficacy, stability, and biocompatibility. Future research should prioritize elucidating molecular interaction dynamics and developing safer, high-affinity counter-ions to overcome current limitations in peptide drug development.

## 1. Introduction

Since the discovery of insulin for diabetes treatment in the 1920s, peptides have emerged as a vital class of pharmaceuticals. Initially, natural insulin was extracted from the pancreas of animals for clinical treatment [1]. With the progress of biotechnology, peptide drugs have developed from natural peptides to synthetic peptides, and synthetic oxytocin has begun to enter the market [2]. To date, the U.S. Food and Drug Administration (FDA) has approved more than 60 peptide drugs, with numerous others undergoing clinical trials [3]. Advances in biotechnology have enabled large-scale production of peptides through genetic engineering and phage display. However, challenges such as poor bioavailability, instability, and rapid metabolism persist [4].

The biological activity of peptides is governed by their spatial conformation and interactions with counter-ions, which electrostatically bind to charged residues on peptide surfaces. Peptides, composed of amino acid residues, possess side chain groups that are either acidic or alkaline, conferring them a net charge. Ions of opposite charge, known as counter-ions, are attracted to these charged residues, forming an electrostatic association. Counter-ions can be either anions or cations, with trifluoroacetic acid (TFA) being one of the most used counter-ions (Figure 1). The presence of counter-ions can significantly influence the spatial conformation and biological activity of peptides, and in some cases, may even induce toxic effects [5]. In recent years, counter-ions have gained attention for their roles in drug delivery systems [6] and reversed-phase high-performance liquid chromatography (RP-HPLC) [7].

This review systematically bridges the critical gap between the fundamental role of counter-ions in modulating peptide structure as well as activity and their applied roles in drug delivery and analytics. While existing reviews have predominantly focused on singular aspects such as the fundamental effects of counter-ions on peptide structure and the function of hydrophobic ion pairing (HIP) in peptide delivery systems, this work provides a comprehensive synthesis that bridges these domains [8,9]. Specifically, this review is scoped to comprehensively cover the types of counter-ions, their effects on peptide structure and biological function, their strategic applications in enhancing peptide drug delivery systems, and the methods for counter-ion separation and determination. By providing a comprehensive analysis of these aspects, this work offers novel insights into the development of peptide-based applications.

Accordingly, this review first outlines a systematic classification of common counter-ions and their fundamental interactions with peptides. It then delves into the critical role of counter-ions in modulating peptide conformation, stability, and biological activity. The discussion subsequently extends to the strategic utilization of counter-ions in advanced drug delivery platforms, with an emphasis on improving membrane permeability and stability. Analytical methodologies for counter-ion separation and quantification are also reviewed. This article concludes with a synthesis of the key findings and a perspective on future research trajectories.

## 2. Types of Counter-Ions

Salification is a widely employed strategy in drug development to modulate bioavailability and solubility, with the choice of counter-ions significantly influencing these properties. Counter-ions are classified as anions or cations based on their charge and further categorized as acidic or alkaline depending on their pKa values (Figure 2). The selection of acidic or alkaline counter-ions is guided by the acid-base properties of the parent drug’s functional groups. Historically, inorganic anions such as chloride, sulfate, and phosphate were predominantly used in drug synthesis. However, organic counter-ions, particularly carboxylates (e.g., acetate, lactate, gluconate, maleate, malate, and citric acid), have gained prominence due to their superior solubility and reduced toxicity. For instance, acetate has largely replaced TFA in peptide drug synthesis, as TFA was found to negatively impact peptide stability and biological activity [10]. These organic counter-ions are now extensively utilized in pharmaceutical formulations, offering enhanced performance and safety profiles.

Cationic counter-ions play a crucial role in drug synthesis and are categorized into inorganic and organic types. Inorganic cationic counter-ions, such as Na^+^, Mg^2+^, K^+^, and Ca^2+^, are widely utilized due to their stability and water solubility. Among these, sodium salts are the most prevalent, exemplified by heparin sodium, which was among the first cationic salts applied in clinical settings. Sodium salts are commonly formed with various anionic raw materials, including phosphate, sulfonate, amide, and enol derivatives. Beyond sodium, other inorganic cations such as potassium, magnesium, calcium, and silver salts are also employed to meet specific drug requirements. In addition to inorganic cations, organic cationic counter-ions, such as tromethamine, meglumine, and erbumine, have been incorporated into FDA-approved drugs [11]. The choice of counter-ions significantly influences the biological properties of peptide drugs, enabling tailored formulations to address diverse therapeutic needs.

## 3. Effects of Counter-Ions on Peptides

### 3.1. Peptide Structure Modulation

The spatial structure of peptides is intrinsically linked to their biological activity, with counter-ions being a key factor influencing peptide conformation and, consequently, their function. For instance, the antimicrobial peptide LL-37 exhibits a disordered structure in aqueous solutions, resulting in diminished antibacterial activity. However, in the presence of counter-ions such as SO_4_^2−^, HCO_3_^−^, and CF_3_CO_2_^−^, LL-37 adopts an α-helical conformation, significantly enhancing its antimicrobial efficacy (Figure 3) [12]. This demonstrates the critical role of counter-ions in modulating peptide secondary structure.

Further evidence supports the broader impact of counter-ions on peptide activity. Further evidence supports the broader impact of counter-ions on peptide activity. Gaussier H et al. [13] demonstrated that TFA^−^ remained persistently associated with the pediocin PA-1 peptide, interfering with structural analysis and inducing subtle conformational perturbations, notably a slight increase in α-helical content. In contrast, Cl^−^ did not exhibit these associative or structure-altering effects, thereby providing a structurally benign alternative for purification. Studies have shown that cyclic peptides [14], beta-casein-(1–25) [15], and laspartomycin [16] are all influenced by counter-ion interactions. For example, SCN^−^ and I^−^ have been found to bind to peptide main chains through hybridization sites involving amide nitrogen and adjacent α-carbon atoms [17]. The molecular mechanisms by which counter-ions direct peptide folding primarily involve charge screening and specific ion-peptide interactions. Charge screening reduces electrostatic repulsion between charged residues on the peptide, thereby facilitating the collapse into stable secondary structures like α-helices [18]. Furthermore, specific ions can engage in direct molecular interactions, such as forming hydrogen bonds with polar backbone or side-chain groups or participating in hydrophobic interactions that stabilize the peptide core [19,20]. These diverse interactions, charge screening, hydrogen bonding, and hydrophobic effects, collectively underscore that counter-ion selection is a critical determinant of peptide activity. Therefore, counter-ions that do not compromise peptide function should be prioritized in drug development.

Furthermore, counter-ions play a pivotal role in the assembly of peptide polymers. Amphiphilic peptides undergo self-assembly when exposed to molecules or counter-ions with opposite charge [21]. The amphiphilic peptides containing acidic amino acids can form nanofibers under acidic conditions or through electrostatic interactions with counter-ions [22]. Calcium ions [23], TFA and formic acid [24] have been shown to influence the dimer formation of peptides or proteins in high-performance liquid chromatography (HPLC). Additionally, sodium dodecyl sulfate (SDS) and sodium chloride (NaCl) have been shown to modulate micelle binding [25]. Counter-ions also significantly impact the packaging of polyoxometalates (POMs), with POMs exhibiting distinct structural states in the presence of counter-ions such as NH_4_^+^, Cu^2+^, and Na^+^ [26]. Collectively, these studies highlight the critical influence of counter-ion charge, type, and coordination mode on peptide assembly, underscoring their importance in the design and development of peptide-based therapeutics.

### 3.2. Positive Effects

The type, size, and coordination of counter-ions significantly influence the bioactivity, water solubility, stability, and bioavailability of peptides. Inorganic counter-ions often outperform organic counter-ions for certain peptide drugs. For instance, MAGE-3, a clinical N-terminal glutamate-containing CTL peptide, presents enhanced stability in its hydrochloride form compared to its acetate form [27]. When anions serve as counter-ions, their size plays a crucial role in the stability of the complex. Larger anions contribute to greater complex stability by forming stronger hydrogen bonds with the functional groups of the receptor and better fitting the host cavity due to their larger ionic radius [28]. A comparative summary of the key distinctions between inorganic and organic counter-ions is provided in Table 1.

Counter-ions also impact the water solubility of compounds through HIP. HIP has been successfully applied to proteins and polynucleotides, where the HIP complex maintains the natural structure and enzyme activity (Figure 4). HIP is instrumental in drug delivery systems, enhancing drug bioavailability [29]. The influence of counter-ions on peptide drugs underscores the importance of selecting the most suitable salt form during novel drug development to maximize therapeutic efficacy. During the screening of suitable counter-ions for the osmotic drug diclofenac (DF), alkaline amino acids such as L-arginine, L-histidine, L-lysine, and their salts were employed as counter-ions. Penetration studies conducted on pig skin revealed that these amino acids could effectively enhance the distribution and permeation of ionic drugs [30]. These findings highlight the potential of amino acids as counter-ions to improve drug delivery and provide valuable insights for the development of transdermal drug delivery systems.

### 3.3. Toxicity

In contrast to the protective effects of certain counter-ions, others may induce unintended toxicological consequences. TFA, a widely employed counter-ion for drug salt formation, has raised significant biocompatibility concerns. For instance, studies reveal that myelin oligodendrocyte glycoprotein (MOG) formulated with TFA and acetic acid induced encephalomyelitis at comparable incidence and severity levels. However, the TFA-containing formulation accelerated disease onset by approximately 5 days compared to controls [31]. Similarly, the TFA salt of the tetra-branched peptide M33 exhibited 5–30% greater cytotoxicity toward normal bronchial epithelial cells than its acetate counterpart [32]. These findings suggest acetate as a safer alternative to TFA in pharmaceutical formulations.

Further compounding its risks, TFA has been shown to stimulate C6 mouse glioma cell proliferation and enhance protein synthesis [33], exacerbate lectin-induced cell agglutination, and suppress osteoblast and articular chondrocyte growth [5] (Figure 5). The underlying mechanisms of TFA’s toxicity are multifaceted, potentially involving the generation of fluorinated metabolites that induce oxidative stress and disrupt mitochondrial function [34,35]. Additionally, TFA adducts metabolites derived from halothane can provoke immune-mediated hepatotoxicity [36], playing a critical role in the pathogenesis of halothane-induced acute liver injury. This occurs as these adducts are recognized as neoantigens by the immune system, triggering an inflammatory response. Collectively, these observations underscore the necessity of rigorous counter-ion evaluation during drug development to mitigate adverse biological effects.

## 4. Selectivity of Peptides Toward Counter-Ions

In addition to the effects of counter-ions on peptide drugs, peptides themselves exhibit distinct affinities for different counter-ions, demonstrating a phenomenon known as ion selectivity. For instance, a synthetic membrane-anchored heptapeptide has been shown to selectively bind chloride ions [37], offering a novel platform for studying chloride ion transport mechanisms [38]. Similarly, cyclic peptide-based anion receptors have demonstrated high affinity for sulfate anions, even in aqueous environments or phosphate buffers. This remarkable selectivity is attributed to the sulfate-binding mechanisms analogous to those of sulfate-binding proteins [39].

Leveraging this ion selectivity, researchers can design drugs capable of isolating target anions from complex solvent environments. Studies have shown that incorporating anion-recognition groups with rigid backbones or specific side chains can significantly enhance both anion affinity and selectivity [40]. These findings underscore the importance of considering peptide-ion selectivity during the development of novel peptides, as it can profoundly influence their functional and therapeutic properties.

## 5. Counter-Ions in Peptide Delivery

Counter-ions play a pivotal role in drug delivery mechanisms by influencing molecular assembly and conformational changes. This relationship provides a versatile platform for modifying drug delivery systems through counter-ion adjustment and the design of HIP. When integrated with other carrier-based strategies, counter-ion modulation holds significant potential to enhance drug targeting and delivery efficiency [41].

For instance, lipid-based nanocarriers—including oil-in-water nanoemulsions, self-emulsifying drug delivery systems (SEDDS), solid lipid nanoparticles (SLNs), nanostructured lipid carriers (NLCs), liposomes, and micelles—can be optimized by incorporating hydrophobic counter-ions. These counter-ions form HIP with the drug, thereby increasing its lipophilicity and facilitating its integration into the lipophilic phase of the carriers [6]. For example, ion pairing semaglutide with ethyl lauroyl arginate (ELA) yielded hydrophobic complexes that conferred enhanced lipophilicity and improved membrane permeability [42], and HIP of tobramycin with sodium docusate significantly increased lipophilicity and enabled stable SEDDS incorporation for oral delivery [43]. This approach not only strengthens drug-carrier compatibility but also improves overall delivery performance.

### 5.1. Nanoemulsion Delivery Systems

Nanoemulsions are composed of stable, nanoscale suspended droplets with high loading capacities and are formulated using safe, biocompatible compounds (Figure 6). These systems offer numerous advantages, including high energy efficiency, scalability for industrial production, robust loading capabilities, and the ability to preserve sensitive or fragile compounds. For example, B. D. da Silva et al. developed sub-100 nm nanoemulsions via ultrasound to significantly enhance the bioactivity of oregano essential oil, carvacrol, and thymol [44], and Fengting, Lei et al. also prepared pH-responsive sodium alginate (SA) hydrogel-coated nanoemulsions to co-deliver CUR and EMO (CUR/EMO NE@SA) to achieve controlled drug release and specifically target macrophages of the colon [45]. The properties of nanoemulsions, such as droplet size and stability, are influenced by factors including the order and rate of compound mixing, surfactant selection, dispersed phase characteristics, and oil properties [46].

Notably, the electrostatic and spatial repulsion effects of counter-ions such as Na^+^ and Cl^−^ have been shown to enhance the stability of nanoemulsions [47]. Due to their excellent biocompatibility and sustainability, nanoemulsion-based delivery systems are widely applicable in the food, cosmetics, and pharmaceutical industries [48]. While effective for encapsulation, nanoemulsions benefit from HIP primarily by enhancing the loading of inherently hydrophilic peptides into the lipid phase, thereby improving encapsulation efficiency and stability.

### 5.2. Self-Emulsifying Drug Delivery System (SEDDs)

The SEDDs are an innovative formulation, typically existing in either solid or liquid states, that comprises three key components: an oil phase, a non-ionic surfactant, and a cosurfactant (Figure 7). A significant advancement in this field has been the incorporation of HIP, formed through counter-ion interactions, which have been shown to enhance the oral bioavailability of peptide drugs while simultaneously providing robust protection against enzymatic degradation. It was reported that the lipophilicity of insulin glargine (IG) was successfully increased via HIP with sodium octadecyl sulfate to enable incorporation into SEDDS [49], and a significant advancement arose in enhancing the oral bioavailability of insulin IG through the innovative use of the polyglycerol/zwitterion-based SEDDS [50]. A landmark study utilizing leuprorelin as a model peptide drug demonstrated the efficacy of self-microemulsifying drug delivery systems in shielding against luminal enzymatic metabolism [51]. This pioneering research marked the first successful integration of HIP into SEDDs for peptide drug delivery, providing compelling evidence of its protective effects against enzymatic degradation. These findings underscore the potential of SEDDs as a transformative platform technology for improving the oral bioavailability of peptide therapeutics.

Further validation of this approach comes from research on oral insulin delivery systems. Innovative mucopermeable formulations incorporating insulin/dimyristoyl phosphatidylglycerol (INS/DMPG) hydrophobic ion pairs have demonstrated dual benefits: effective protection against degradation by intestinal enzymes (specifically trypsin and α-chymotrypsin) and enhancement of the initial burst release of insulin [52]. These results suggest a promising pathway for the development of effective oral insulin administration strategies. The application of HIP technology has been further extended to desmopressin SEDDs formulations, which have shown significant protection against inactivation by glutathione and α-chymotrypsin in vitro [53].

Among the lipid-based systems discussed, SEDDs derive the most significant and direct benefit from HIP. The technology is pivotal for SEDDs. The formation of HIP is driven by a combination of electrostatic interactions and the hydrophobic effect. The electrostatic attraction between the charged peptide and the oppositely charged hydrophobic counter-ion initiates the pair formation. Subsequently, the system’s overall free energy is lowered by minimizing the unfavorable contact between the hydrophobic moieties of the counter-ion and the aqueous environment. This process results in a net increase in the complex’s lipophilicity, which directly facilitates partitioning into and encapsulation within the lipophilic matrices of nanocarriers, leading to the enhanced oral bioavailability observed in these systems [54].

### 5.3. Solid Lipid Nanoparticles System (SLNs)

SLNs represent an advanced class of nanocarriers with a particle size ranging from 10 to 1000 nm. These nanoparticles are composed of a solid lipid core, which can be derived from natural or synthetic lipids such as lecithin and triacylglycerol, and are capable of encapsulating or adsorbing drugs within their lipid matrix. As a next-generation drug delivery system, SLNs leverage the unique advantages of solid lipids, including low toxicity, excellent biocompatibility, and biodegradability, making them an ideal platform for therapeutic applications (Figure 8).

The incorporation of HIP into SLNs has emerged as a promising strategy for developing in vitro sustained-release systems for peptide drugs [55]. This approach has been successfully demonstrated using an HIP complex formed between octreotide acetate and dextran sulfate sodium (DSS), which validated the feasibility of this strategy [56]. The HIP-enhanced SLN system exhibits a stable and safe blood drug concentration profile, coupled with high in vivo bioavailability. Key advantages of this system include high encapsulation efficiency, minimal initial burst release, and a stable drug release mechanism.

The primary advantage of integrating HIP into SLNs lies in achieving superior controlled release kinetics. The solid lipid matrix provides a robust barrier that, combined with the hydrophobized peptide, minimizes the initial burst release and enables sustained drug release, which is a distinct advantage over the faster-release profiles often seen in nanoemulsions and SEDDs.

### 5.4. Nanostructured Lipid Carriers System (NLCs)

Nanoprecipitation of active pharmaceutical ingredients to form nanocarriers has emerged as a promising technique for developing formulations with enhanced stability and biological efficacy. It was indicated that the NLCs significantly enhanced the permeation and retention of quercetin within the skin layers [57], and charge-converting nanostructured lipid carriers containing a cell-penetrating peptide could enhance cellular uptake [58]. However, the application of nanoprecipitation technology to highly soluble peptide therapeutics remains unproven. To address this limitation, researchers have proposed a novel strategy involving the modification of peptide solubility through the formation of hydrophobic ion pairs with counter-ions, thereby rendering the peptides compatible with nanoprecipitation techniques [59].

NLCs have demonstrated significant potential in improving the oral bioavailability of peptide drugs, offering a viable solution to overcome the inherent challenges associated with oral peptide delivery (Figure 8). This system is capable of encapsulating lipidated peptides without compromising the structural integrity or biological activity of the therapeutic agent. Furthermore, the versatility of NLC technology allows for the adaptation of processing methods to accommodate the intrinsic properties of various peptides, enabling the large-scale production of peptide-loaded nanoparticles. Notably, NLCs have been shown to provide robust protection against protease-induced degradation and enhance the transport of peptides across epithelial barriers [60].

NLCs benefit from HIP in a manner similar to SLNs but with enhanced performance. The imperfect solid lipid matrix of NLCs offers higher drug loading capacity and reduces drug expulsion during storage. When used with HIP-complexed peptides, NLCs achieve an optimal balance between high loading efficiency and superior sustained-release profiles, making them particularly suited for long-acting peptide delivery applications.

### 5.5. Liposomes and Micelles

Liposome-based drug delivery systems offer precise control over the temporal and spatial release of therapeutic agents through the use of internal and external triggers. These triggers include pH, enzymatic activity, redox conditions, temperature, magnetic fields, electromagnetic waves, and ultrasound, enabling targeted and stimuli-responsive drug release [61]. The incorporation of counter-ions into liposome formulations has emerged as a promising strategy for enhancing the delivery of peptide drugs [62]. For instance, the strategic incorporation of multivalent counterions, such as Ca^2+^, induces the structural transformation of anionic liposomes into stable, solid nanocochleates, which effectively encapsulate and protect peptide-based therapeutics, thereby enhancing their stability and enabling efficient cellular delivery through membrane fusion mechanisms [63].

Counter-ions play a critical role in optimizing peptide drug delivery systems by forming hydrophobic ion pairs, which not only improve the stability and bioavailability of peptide therapeutics but also provide protection against enzymatic degradation.

For liposomes, HIP serves to dramatically increase the encapsulation efficiency of hydrophilic peptide drugs within the aqueous core or at the lipid bilayer interface, which is a key challenge for this platform. Furthermore, certain counter-ions can actively promote liposome fusion with cell membranes, thereby enhancing cellular uptake. Micellar systems, which rely on the assembly of amphiphilic molecules, benefit from HIP as the primary mechanism to integrate peptide drugs into the hydrophobic micelle core. This integration is crucial for stabilizing the peptide and achieving satisfactory drug loading in these nanocarriers.

### 5.6. Comparative Analysis of Delivery Platforms

SEDDs are the most reliant on HIP, and technology is fundamental to their function in oral peptide delivery, enabling both drug incorporation and self-emulsification. Nanoemulsions utilize HIP mainly to improve initial drug loading. In contrast, SLNs and NLCs, as solid-core particles, leverage HIP primarily to fine-tune release kinetics; SLNs excel in providing a more rigid framework for sustained release, while NLCs offer greater versatility for loading and delivering complex peptides.

Notably, liposomes and micelles represent a distinct category where HIP is employed to solve the critical challenge of loading water-soluble peptides into lipid-based structures. For liposomes, the focus is on enhancing encapsulation efficiency and promoting cellular interactions, while for micelles, HIP is fundamental to creating the requisite drug-polymer affinity for stable encapsulation.

## 6. Counter-Ions in Chromatography

The strategic selection of counter-ions as mobile phase additives is critical for optimizing the chromatographic behavior of peptides, directly influencing retention, peak shape, and detection sensitivity. In reversed-phase liquid chromatography (RPLC), hydrophobic counter-ions like TFA dynamically mask charged sites on the peptide through electrostatic interactions, forming ion pairs that increase their effective hydrophobicity and thereby modulating retention time and enhancing separation resolution [64,65]. This effect can eliminate peak splitting and improve peak shape by reducing undesirable secondary interactions with silanol groups on the stationary phase [36].

However, the choice of counter-ion entails significant trade-offs. While TFA is highly effective for UV-based detection, it is notorious for causing severe ion suppression in electrospray ionization mass spectrometry (ESI-MS) due to its tendency to form gas-phase ion pairs that compete for charge, thereby diminishing analyte signal [7]. More hydrophilic alternatives like formic acid (FA) offer better MS compatibility but may provide inferior chromatographic resolution for some peptides. Beyond retention and detection, certain counter-ions, including TFA and FA, can influence peptide conformation and oligomerization during analysis, potentially inducing molten globule states or enhancing protein dimerization, which can further complicate chromatographic outcomes [66,67,68].

## 7. Determination and Separation of Counter-Ions

The determination and separation of counter-ions can be achieved through a variety of analytical techniques, each with its specific principles and applications. A comparative overview of these methods, including electrophoresis (CE), ion chromatography (IC), isotachophoresis (ITP), and LC-MS/LC-ELSD, is provided in Table 2. The following sections detail the most commonly employed and emerging techniques.

### 7.1. Determination Methods

The determination of counter-ions can be achieved through various analytical techniques, with capillary CE [69], IC and ITP being the most commonly employed methods.

CE is a versatile technique capable of analyzing a wide range of analytes, from small inorganic ions to high-molecular-weight biomolecules, particles, and even intact cells. Its ability to quantitatively measure target molecules with high precision makes it a valuable tool in analytical chemistry [70]. Additionally, ultraviolet (UV) spectrophotometry has been utilized to detect inorganic anions and cations in thin-tube electrophoresis systems [71]. A notable advantage of CE is its ability to simultaneously separate cations and anions through the strategic manipulation of electrolytes and sample matrices [72]. The technique has been extensively documented in scientific literature, demonstrating its reliability and broad applicability across diverse fields [73].

IC, a specialized form of HPLC, has become a cornerstone in ion analysis due to its ability to detect nearly all types of ions. Its widespread use is attributed to its sensitivity and versatility [74]. For instance, trace amounts of TFA can be isolated from complex matrices containing excess chloride ions, phosphates, and other anions using high-capacity anion exchange columns, often without the need for extensive sample preparation [75]. Furthermore, advanced ion chromatography systems equipped with conductivity detection have been developed for the efficient and sensitive quantification of inorganic cations (e.g., Na^+^, NH_4_^+^, K^+^, Mg^2+^, Ca^2+^) and anions (e.g., chloride, acetate, lactate) in industrial concentrates [76].

ITP, a powerful electrophoretic technique derived from capillary electrophoresis, offers unique capabilities for sample pre-concentration, separation, purification, and mixing, as well as for controlling and accelerating chemical reactions. Its application in solving complex analytical problems has garnered increasing attention. When coupled with mass spectrometry (MS), ITP enhances analytical efficiency and sensitivity [77]. However, challenges remain in the determination of chloride ions, highlighting the need for further technical refinement [78].

Beyond these traditional methods, emerging techniques are expanding the capabilities of ion detection. For example, voltammetry has been employed to quantify iron (II) and iron (III) in pharmaceutical iron–polysaccharide complexes [79]. Liquid chromatography–tandem mass spectrometry (LC-MS/MS) enables the quantitative determination of acetate in pharmaceutical peptide preparations while simultaneously monitoring peptide content during manufacturing and in final products [80]. Additionally, innovative approaches, such as establishing calibration curves using frozen hydration buffers with known ionic strength, have been developed to determine Na^+^ concentrations in polyacrylic acid (PAA) microgels [81]. A recent advancement in evaporative light scattering detection (ELSD) coupled with mixed-mode HPLC allows for the simultaneous quantification of 14 positively and negatively charged counter-ions within 30 min, demonstrating excellent precision and accuracy [82].

As analytical technologies continue to evolve, the detection and quantification of ions are becoming increasingly efficient and accessible. The limitations and blind spots in ion analysis are gradually diminishing, paving the way for more comprehensive and reliable analytical solutions.

### 7.2. Separation Methods

The purification of protein and peptide drugs necessitates the separation of counter-ions following their detection. A variety of methods are available for this purpose, ranging from classical techniques to innovative approaches that enhance separation efficiency and sample purity.

The ion exchange resin method and RP-HPLC remain the most widely used classical techniques for counter-ion separation. However, advancements in separation technology have introduced more efficient methods. For instance, a novel mixed-mode reversed-phase/weak anion exchange stationary phase has demonstrated superior selectivity and higher sample loading capacity for the separation and purification of peptides, significantly improving sample yield [83]. Another innovative approach involves the use of a single silica column based on hydrophilic interaction chromatography (HILIC) coupled with ELSD. This method has proven effective for the simultaneous separation and detection of counter-ions, offering a streamlined and efficient solution [84].

Recent developments have also highlighted the potential of acid freeze-drying methods for counter-ion separation. Specifically, hydrochloric acid or other acids have been shown to facilitate reverse ion exchange of peptides, with enhanced efficiency observed in organic solutions [85].

These emerging technologies not only improve the separation and purification of counter-ions but also contribute to higher yields and better sample quality. As the field continues to evolve, the integration of these advanced methods is expected to further optimize the purification processes for protein and peptide therapeutics.

## 8. Conclusions

The exploration and development of peptide drugs remain a dynamic and evolving field, with numerous research challenges that are yet to be addressed. Overcoming these challenges will mark a significant milestone in the advancement of peptide-based therapeutics. Among the various factors influencing the biological activity of peptide drugs, counter-ions play a pivotal role, contributing to their development in multiple ways. A deeper understanding of the specific effects of counter-ions on peptides could enable the synthesis of peptide drugs with enhanced biological activity and therapeutic efficacy.

However, the precise mechanisms by which counter-ions influence peptide behavior are not yet fully elucidated, leaving a critical gap in our knowledge. Addressing this gap through systematic research will not only clarify the role of counter-ions but also pave the way for innovative strategies in peptide drug design and optimization. As the field progresses, a more comprehensive understanding of counter-ion interactions will undoubtedly contribute to the development of next-generation peptide therapeutics with improved performance and clinical outcomes.

## Figures and Tables

**Figure 1 biomolecules-15-01567-f001:**
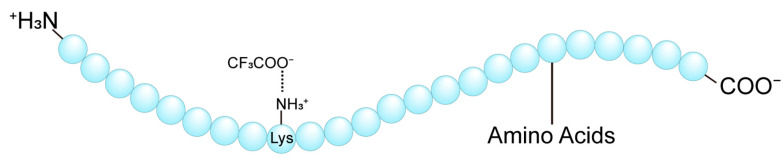
Schematic diagram of electrostatic association between TFA and polypeptide chain. In solution, TFA dissociates into CF_3_COO^−^ and H^+^.

**Figure 2 biomolecules-15-01567-f002:**
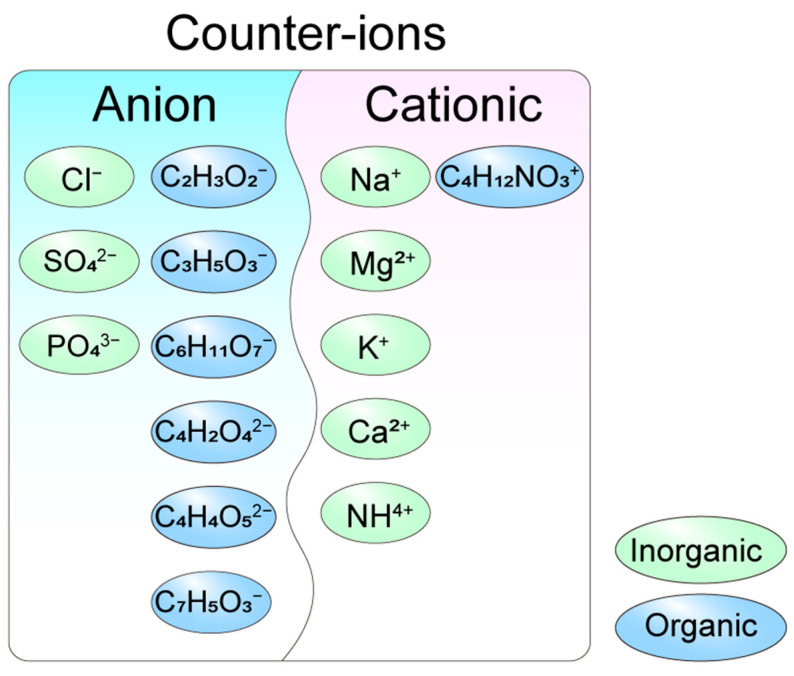
Types of counter-ions in the acid radical ion form. Inorganic anion counter-ions: Cl^−^: chloride, SO_4_^2−^: sulfate, and PO_4_^3−^: phosphate. Organic anion counter-ions: C_2_H_3_O_2_^−^: acetate, C_3_H_5_O_3_^−^: lactate, C_6_H_11_O_7_^−^: gluconate, C_4_H_2_O_4_^2−^: succinate, C_4_H_4_O_5_^2−^: tartrate and C_7_H_5_O_3_^−^: salicylate. Inorganic cationic counter-ions: Na^+^: sodium ion, Mg^2+^: magnesium ion, K^+^: potassium ion, Ca^2+^: calcium ion, and NH_4_^+^: ammonium ion. Organic cationic counter-ion: C_4_H_12_NO_3_^+^: choline ion.

**Figure 3 biomolecules-15-01567-f003:**
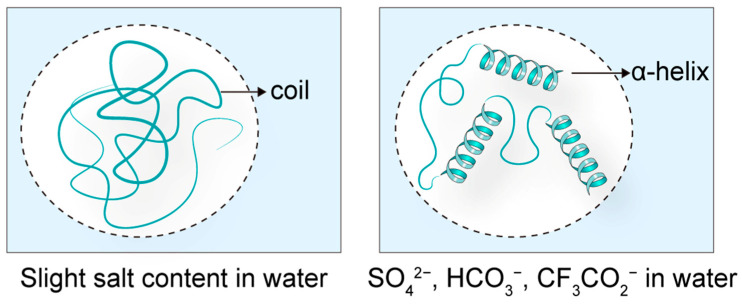
The effects of counter-ions on peptide structure. When the antimicrobial peptide LL-37 is in a solution close to water, its structure is in a disordered state. When placed in solutions containing SO_4_^2^¯, HCO_3_¯, and CF_3_CO_2_¯, it was induced to form an α-helix structure.

**Figure 4 biomolecules-15-01567-f004:**
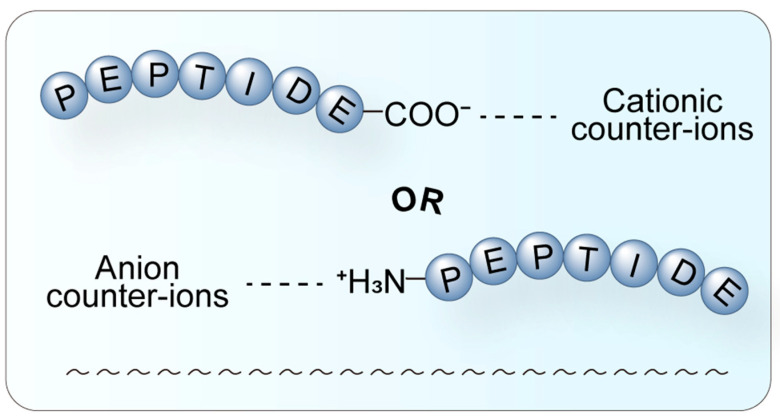
Schematic diagram of the HIP. HIP is a technique where hydrophilic molecules form complexes with hydrophobic counter-ions through electrostatic interaction. A hydrophilic peptide (blue) interacts with a hydrophobic counter-ion through electrostatic attraction (dashed line). This process effectively masks the surface charge of the parent molecule, significantly increasing its overall lipophilicity, thereby facilitating encapsulation into lipid nanocarriers and enhancing membrane permeability and oral bioavailability.

**Figure 5 biomolecules-15-01567-f005:**
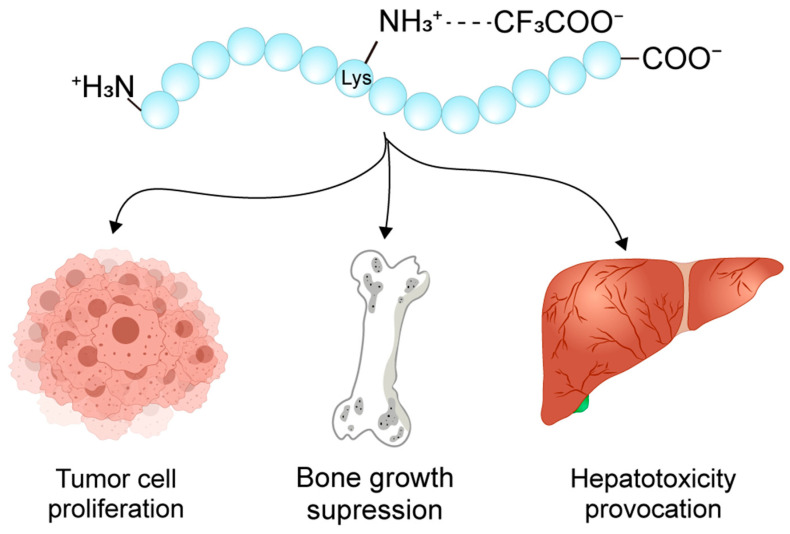
Schematic diagram of the peptide toxicity in the TFA salt form. TFA has been shown to contribute to glioma cell proliferation, osteoblast and articular chondrocyte damage, as well as acute liver injury.

**Figure 6 biomolecules-15-01567-f006:**
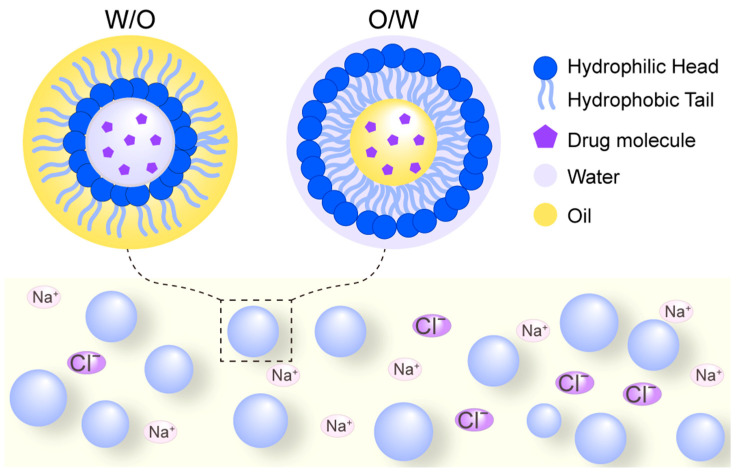
Schematic diagram of nanoemulsion delivery systems. W/O denotes a water-in-oil emulsion where aqueous droplets are dispersed within a continuous oil phase, while O/W represents an oil-in-water emulsion where oil droplets are dispersed in a continuous aqueous phase. These systems are composed of safe, biocompatible compounds including oil phases, aqueous phases, and surfactants. The incorporation of counter-ions can enhance stability through electrostatic and spatial repulsion effects.

**Figure 7 biomolecules-15-01567-f007:**
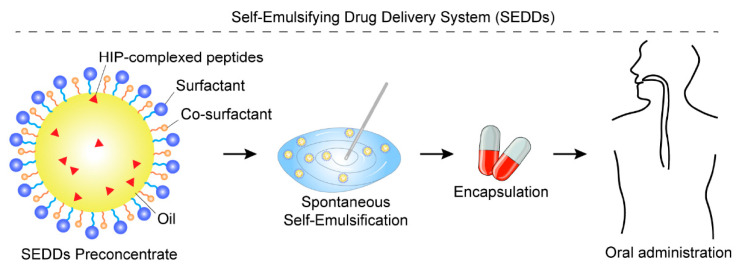
Schematic illustration of SEDDs enhanced by HIP. The SEDD preconcentrate consists of an oil phase, a surfactant, a co-surfactant, and peptide drugs that have been hydrophobically modified via HIP complexation with counter-ions. Upon aqueous dilution and gentle agitation, the preconcentrate rapidly forms a fine nanoemulsion, which encapsulates the HIP-complexed peptides, thereby protecting them from enzymatic degradation and enhancing their oral absorption.

**Figure 8 biomolecules-15-01567-f008:**
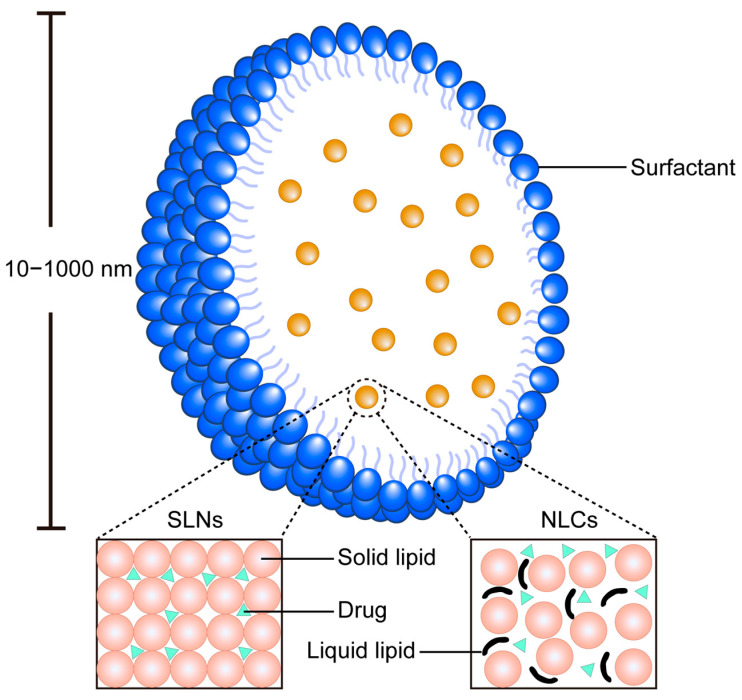
Schematic diagram of the SLN system and NLCs. SLNs comprise a solid lipid core stabilized by surfactants, providing a rigid matrix for sustained drug release. NLCs feature a blended solid–liquid lipid matrix that creates structural imperfections, enabling higher drug loading capacity.

**Table 1 biomolecules-15-01567-t001:** Comparison of inorganic and organic counter-ions for peptide drugs.

Feature	Inorganic Counter-Ions	Organic Counter-Ions
Key Characteristics	Small size, high charge density, strong electrostatic	Large size, functional groups, specific interactions
Typical Effects on Peptides	Enhances stability, effective charge screening	Modulates solubility via HIP and enhances permeation
Advantages	High stability, cost-effective, well-established	Tunable properties, functional, targets specific delivery
Disadvantages	Limited tunability, basic functionality	Complex synthesis, higher cost, potential conformational risk

**Table 2 biomolecules-15-01567-t002:** Comparison of analytical techniques for counter-ion determination.

Technique	Applications	Advantages	Limitations
CE	Broad ion analysis	High efficiency, minimal sample	Moderate sensitivity
IC	Universal ion detection	High sensitivity, robust	Specific columns required
ITP	Sample preconcentration	Excellent focusing capability	Method complexity
LC-MS/LC-ELSD	Specific ion quantification	High specificity, universal	Costly instrumentation

## Data Availability

Not applicable.

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
