# Peer review of "The Effects of Counter-Ions on Peptide Structure, Activity, and Applications"

_biomolecules, 2025, doi:10.3390/biom15111567_

Round 1
Reviewer 1 Report
Comments and Suggestions for Authors
This review manuscript presents a well-organized and comprehensive overview of the multifaceted roles of counter-ions in peptide chemistry. The topic is timely and relevant, especially given the increasing number of peptide-based therapeutics and the critical importance of counter-ion effects on solubility, stability, and biological performance. The iThenticate score (only 18%) is very nice and the topic is surely within the scope of the SI: Antimicrobial Peptides in Nature: Inspiration for Rationally Designed Antimicrobials. However, the manuscript in its current form still requires considerable improvement before being ready for publication. The main issues include:
Detailed comments:
There is insufficient coverage of recent literature (2021–2025) on counter-ion effects and peptide delivery systems; the reference list stops around 2020.
The introduction should more clearly define the scope of the review and specify the novelty compared to previous reviews.
I suggest adding a final paragraph to introduction, outlining the manuscript’s structure.
The LL-37 example is well-chosen, but subsequent examples are presented in a list-like style. Consider summarizing the molecular mechanisms of ion-specific folding (e.g., through charge screening, hydrogen bonding, etc.).
Lines 136–157: The section could benefit from a short comparative table summarizing inorganic vs. organic counter-ions and their typical effects.
Lines 161–179: Consider adding more mechanistic context about TFA’s toxicity (e.g., fluorinated metabolites, oxidative stress).
The discussion of nanoemulsions, SEDDs, SLNs, and NLCs is well-structured but highly descriptive. Include some comparative analysis (e.g., which platform benefits most from hydrophobic ion pairing).
Line 346, the authors should mention capillary electrophoresis and cite Grodner, B., & Jelińska, M. (2023). The method of direct determination of manganese in fresh parsley leaves and roots using capillary electrophoresis method. Prospects in Pharmaceutical Sciences, 21(1), 5–8. https://doi.org/10.56782/pps.125
Lines 313–341: This section reads like a methods overview. It could be condensed and focused on how counter-ion selection directly affects chromatographic outcomes for peptides (e.g., retention shifts, ion suppression effects).
Author Response
Reviewer 1
This review manuscript presents a well-organized and comprehensive overview of the multifaceted roles of counter-ions in peptide chemistry. The topic is timely and relevant, especially given the increasing number of peptide-based therapeutics and the critical importance of counter-ion effects on solubility, stability, and biological performance. The iThenticate score (only 18%) is very nice and the topic is surely within the scope of the SI: Antimicrobial Peptides in Nature: Inspiration for Rationally Designed Antimicrobials. However, the manuscript in its current form still requires considerable improvement before being ready for publication.
Response: Thank you for your positive assessment of our manuscript and your encouraging comments regarding its relevance and originality. We greatly appreciate your time and the constructive feedback. We have given careful consideration to all the comments and have undertaken a comprehensive revision of the manuscript to address them. Our detailed, point-by-point responses to each specific comment are provided below. We believe the manuscript has been significantly strengthened through this revision process and hope it now meets the journal's high standards for publication.
Comment 1: There is insufficient coverage of recent literature (2021–2025) on counter-ion effects and peptide delivery systems; the reference list stops around 2020.
Response 1: Thank you for this important observation. We fully agree that incorporating recent literature is crucial for contextualizing our work. In response to your comment, we have conducted a comprehensive review and updated the manuscript by adding recent publications. A total of 19 new references has been incorporated, 11 of which were published between 2021 and 2025. These additions—which include references [3], [8], [9], [18], [19], [20], [34], [35], [42], [43], [44], [45], [49], [50], [57], [58], [63], [65], and [69]—are anticipated to enhance the discussion across multiple sections. It is expected that this thorough revision has addressed your concern and significantly improved the timeliness of the review.
Comment 2: The introduction should more clearly define the scope of the review and specify the novelty compared to previous reviews.
Response 2: Thank you for your valuable comment regarding the need to better define the scope and novelty of our review. We agree that this clarification strengthens the introduction. We have revised the introduction to explicitly state the novelty of our work. As highlighted in the revised manuscript, our review provides a unique, integrated perspective by systematically bridging the fundamental role of counter-ions with their applied roles in drug delivery and analytics, an approach we identify as being often treated in isolation in previous literature.
The revised introduction text clearly defining the scope and specifying the novelty of this review was at line 68-78: “This review systematically bridges the critical gap between the fundamental role of counter-ions in modulating peptide structure as well as activity and their applied roles in drug delivery and analytics. While existing reviews have predominantly focused on singular aspects such as the fundamental effects of counter-ions on peptide structure and function of hydrophobic ion pairing (HIP) in peptide delivery systems while this work provides a comprehensive synthesis that bridges these domains [8,9]. Specifically, this review is scoped to comprehensively cover the types of counter-ions, their effects on peptide structure and biological function, their strategic applications in enhancing pep-tide drug delivery systems, and the methods for counter-ion separation and determination. By providing a comprehensive analysis of these aspects, this work offers novel in-sights into the development of peptide-based applications”.
Comment 3: I suggest adding a final paragraph to introduction, outlining the manuscript’s structure.
Response 3: Thank you for this insightful suggestion. We agree that outlining the manuscript’s structure will enhance its clarity and provide a valuable roadmap for the reader. We have added a new final paragraph to the introduction as you recommended. This paragraph explicitly outlines the logical flow and content of the subsequent sections, from the classification and fundamental effects of counter-ions, through their applications in drug delivery, to analytical methods and the concluding perspective. We believe this addition significantly improves the manuscript’s organization and readability, and we are grateful for your constructive guidance.
The added final paragraph in introduction was at line 79-86: “Accordingly, this review first outlines a systematic classification of common counter-ions and their fundamental interactions with peptides. It then delves into the critical role of counter-ions in modulating peptide conformation, stability, and biological activity. The discussion subsequently extends to the strategic utilization of counter-ions in advanced drug delivery platforms, with an emphasis on improving membrane permeability and stability. Analytical methodologies for counter-ion separation and quantification are also reviewed. The article concludes with a synthesis of the key findings and a perspective on future research trajectories”.
Comment 4: The LL-37 example is well-chosen, but subsequent examples are presented in a list-like style. Consider summarizing the molecular mechanisms of ion-specific folding (e.g., through charge screening, hydrogen bonding, etc.).
Response 4: Thank you for your professional suggestion. We have thoroughly revised the paragraph. Specifically, we have added a concise summary of the key molecular mechanisms beyond the enumeration of peptide examples, including charge screening and specific interactions such as hydrogen bonding and hydrophobic effects. The example of SCN⁻ and I⁻ binding was kept to illustrate a specific coordination mode.
The revised text to summarize the molecular mechanisms of ion-specific folding was at line 141-151: “The molecular mechanisms by which counter-ions direct peptide folding primarily in-volve charge screening and specific ion-peptide interactions. Charge screening reduces electrostatic repulsion between charged residues on the peptide, thereby facilitating the collapse into stable secondary structures like α-helices [18]. Furthermore, specific ions can engage in direct molecular interactions, such as forming hydrogen bonds with polar backbone or side-chain groups, or participating in hydrophobic interactions that stabilize the peptide core [19,20]. These diverse interactions, charge screening, hydrogen bonding, and hydrophobic effects, collectively underscore that counter-ion selection is a critical determinant of peptide activity. Therefore, counter-ions that do not compromise peptide function should be prioritized in drug development”.
Comment 5: Lines 136–157: The section could benefit from a short comparative table summarizing inorganic vs. organic counter-ions and their typical effects.
Response 5: Thank you for your suggestion. We agree that a comparative table would enhance the clarity and accessibility of the information presented in the section on counter-ion types. As recommended, we have added a short comparative table (Table 1) summarizing inorganic vs. organic counter-ions and their typical effects. The table has been incorporated into the section (line 176), and it is now referenced in the text in line 173-175: “A comparative summary of the key distinctions between inorganic and organic counter-ions is provided in Table 1”. We believe this addition enhances the clarity and readability of the section.
Comment 6: Lines 161–179: Consider adding more mechanistic context about TFA’s toxicity (e.g., fluorinated metabolites, oxidative stress).
Response 6: Thank you for your professional suggestion to provide deeper mechanistic context regarding TFA’s toxicity. We have added a sentence outlining the potential mechanisms, including the role of fluorinated metabolites in inducing oxidative stress and disrupting mitochondrial function, to explain the observed cellular toxicities. Furthermore, we have elaborated on how TFA-derived adducts act as neoantigens to trigger the immune-mediated hepatotoxicity.
The text discussing the underlying mechanisms of TFA’s toxicity has been improved to “The underlying mechanisms of TFA’s toxicity are multifaceted, potentially involving the generation of fluorinated metabolites that induce oxidative stress and disrupt mitochondrial function [34,35]. Additionally, TFA adducts metabolites derived from halothane can provoke immune-mediated hepatotoxicity [29], playing a critical role in the pathogenesis of halothane-induced acute liver injury. This occurs as these adducts are recognized as neoantigens by the immune system, triggering an inflammatory response” at line 209-217.
Comment 7: The discussion of nanoemulsions, SEDDs, SLNs, and NLCs is well-structured but highly descriptive. Include some comparative analysis (e.g., which platform benefits most from hydrophobic ion pairing).
Response 7: Thank you for this insightful comment regarding the need for a comparative analysis of the delivery platforms. We agree that moving beyond a descriptive account to a more analytical perspective significantly strengthens the discussion. In response, we have revised the respective sections for nanoemulsions, SEDDs, SLNs, NLCs, liposomes, and micelles. Specifically, we have added a concluding sentence or short paragraph to each subsection that summarizes its unique relationship with and benefit from hydrophobic ion pairing (HIP) technology. Furthermore, as the core of our response, we have incorporated a new subsection 5.6 “Comparative Analysis of Delivery Platforms”. This new subsection synthesizes the information and explicitly addresses your request by analyzing which platform benefits most from HIP (concluding that SEDDs are the most reliant), and delineates the distinct functional advantages HIP confers to each system, including the role of HIP in enhancing the encapsulation and cellular delivery efficiency of liposomes and micelles.
The last paragraph of chapter 5.1 Nanoemulsion Delivery Systems has been improved to “Notably, the electrostatic and spatial repulsion effects of counter-ions such as Na⁺ and Cl⁻ have been shown to enhance the stability of nanoemulsions [47]. Due to their excellent biocompatibility and sustainability, nanoemulsion-based delivery systems are widely applicable in the food, cosmetics, and pharmaceutical industries [48]. While effective for encapsulation, nanoemulsions benefit from HIP primarily by enhancing the loading of inherently hydrophilic peptides into the lipid phase, thereby improving encapsulation efficiency and stability” at line 267-273.
The last paragraph of 5.2 Self-Emulsifying Drug Delivery System has been improved to: “Among the lipid-based systems discussed, SEDDs derive the most significant and direct benefit from HIP. The technology is pivotal for SEDDs. The formation of HIP is driven by a combination of electrostatic interactions and the hydrophobic effect. The electrostatic attraction between the charged peptide and the oppositely charged hydro-phobic counter-ion initiates the pair formation. Subsequently, the system’s overall free energy is lowered by minimizing the unfavorable contact between the hydrophobic moieties of the counter-ion and the aqueous environment. This process results in a net in-crease in the complex’s lipophilicity which directly facilitates partitioning into and en-capsulation within the lipophilic matrices of nanocarriers, leading to the enhanced oral bioavailability observed in these systems [54]” at line 307-316.
The last paragraph of 5.3 Solid Lipid Nanoparticles system has been improved to “The primary advantage of integrating HIP into SLNs lies in achieving superior controlled release kinetics. The solid lipid matrix provides a robust barrier that, combined with the hydrophobized peptide, minimizes the initial burst release and enables sustained drug release, which is a distinct advantage over the faster-release profiles often seen in nanoemulsions and SEDDs” at line 338-342.
The last paragraph of 5.4 Nanostructured Lipid Carriers System has been improved to “NLCs benefit from HIP in a manner similar to SLNs but with enhanced performance. The imperfect solid lipid matrix of NLCs offers higher drug loading capacity and reduces drug expulsion during storage. When used with HIP-complexed peptides, NLCs achieve an optimal balance between high loading efficiency and superior sustained-release profiles, making them particularly suited for long-acting peptide delivery applications” at line 363-367.
The last paragraph of 5.5 Liposomes and Micelles has been improved to “For liposomes, HIP serves to dramatically increase the encapsulation efficiency of hydrophilic peptide drugs within the aqueous core or at the lipid bilayer interface, which is a key challenge for this platform. Furthermore, certain counter-ions can actively promote liposome fusion with cell membranes, thereby enhancing cellular uptake. Micellar systems, which rely on the assembly of amphiphilic molecules, benefit from HIP as the primary mechanism to integrate peptide drugs into the hydrophobic micelle core. This integration is crucial for stabilizing the peptide and achieving satisfactory drug loading in these nanocarriers” at line 387-394.
The chapter 5.6 Comparative Analysis of Delivery Platforms has been added at line 395-406: “SEDDs are the most reliant on HIP, as the technology is fundamental to their function in oral peptide delivery, enabling both drug incorporation and self-emulsification. Nanoemulsions utilize HIP mainly to improve initial drug loading. In contrast, SLNs and NLCs, as solid-core particles, leverage HIP primarily to fine-tune release kinetics; SLNs excel in providing a more rigid framework for sustained release, while NLCs offer greater versatility for loading and delivering complex peptides.
Notably, liposomes and micelles represent a distinct category where HIP is employed to solve the critical challenge of loading water-soluble peptides into lipid-based structures. For liposomes, the focus is on enhancing encapsulation efficiency and promoting cellular interactions, while for micelles, HIP is fundamental to creating the requisite drug-polymer affinity for stable encapsulation.
The choice of platform should therefore be guided by the specific therapeutic goal: SEDDs for maximizing oral absorption, nanoemulsions for rapid delivery, SLNs/NLCs for controlled, long-acting therapy, and liposomes/micelles for targeted delivery and enhanced cellular uptake where their functional surfaces and triggers can be fully leveraged”.
Comment 8: Line 346, the authors should mention capillary electrophoresis and cite Grodner, B., & Jelińska, M. (2023). The method of direct determination of manganese in fresh parsley leaves and roots using capillary electrophoresis method. Prospects in Pharmaceutical Sciences, 21(1), 5–8.
Response 8: Thank you for this suggestion. As recommended, we have now explicitly mentioned capillary electrophoresis at line 436 by citing the reference by Grodner & Jelińska (2023) in the revised manuscript as the reference [69].
Comment 9: Lines 313–341: This section reads like a methods overview. It could be condensed and focused on how counter-ion selection directly affects chromatographic outcomes for peptides (e.g., retention shifts, ion suppression effects).
Response 9: Thank you for your suggestion. We agree that the section would benefit from a more focused discussion on the direct impact of counter-ion selection. In response, we have thoroughly condensed and rewritten the paragraph in chapter 6. Counter-Ions in Chromatography. The revised text now explicitly centers on how counter-ion choice directly dictates chromatographic outcomes for peptides, with a specific emphasis on the mechanism of retention time modulation, the critical trade-off between chromatographic resolution and MS detection sensitivity, and the potential impact on peptide conformation. We have removed the general methodological overview and replaced it with this cause-and-effect analysis, which we believe significantly strengthens the analytical depth of this section.
The whole text at line 408-424 has been improved to “The strategic selection of counter-ions as mobile phase additives is critical for optimizing the chromatographic behavior of peptides, directly influencing retention, peak shape, and detection sensitivity. In reversed-phase liquid chromatography (RPLC), hydrophobic counter-ions like TFA dynamically mask charged sites on the peptide through electrostatic interactions, forming ion pairs that increase their effective hydrophobicity and thereby modulating retention time and enhancing separation resolution [64,65]. This effect can eliminate peak splitting and improve peak shape by reducing undesirable secondary interactions with silanol groups on the stationary phase [36].
However, the choice of counter-ion entails significant trade-offs. While TFA is highly effective for UV-based detection, it is notorious for causing severe ion suppression in electrospray ionization mass spectrometry (ESI-MS) due to its tendency to form gas-phase ion pairs that compete for charge, thereby diminishing analyte signal [7]. More hydrophilic alternatives like formic acid (FA) offer better MS compatibility but may provide inferior chromatographic resolution for some peptides. Beyond retention and detection, certain counter-ions, including TFA and FA, can influence peptide conformation and oligomerization during analysis, potentially inducing molten globule states or enhancing protein dimerization, which can further complicate chromatographic out-comes [66-68]”.
Reviewer 2 Report
Comments and Suggestions for Authors
The manuscript from Ying Liu et al. is a review dealing with the role that counter-ions have on the structure, activity and applications of peptides. This information is key to optimizing their use as pharmacological drugs.
The manuscript is interesting and well-structured, covering the types of counter-ions and their effects on the structure, stability, and toxicity of peptides, as well as their impact on the systems used for peptide delivery or the techniques employed for their identification or separation. However, there are some issues that the author should address before its consideration for publication.
- Although the review is very informative, it is generally too descriptive, so it lacks depth in explaining why and how certain counter-ions work, and also fails to provide more detailed examples of their uses.
- The figure captions are very short, so it is necessary to provide more information about what is described in each one. For example, in figure 2, add the names of the counter-ions shown. In figure 4, give the meaning of HIP and add the information that counter-ions are hydrophobic molecules. In figure 6, add the meaning of W/O and O/W and more details about the nanoemulsion system shown in the figure. The same for figure 7.
- It should be convenient to add a figure describing the other delivery systems mentioned (slns, seedds, nlcs).
- In section 7 there are no figures or schemes, but it would be convenient to add at least one to better understand this point.
- Regarding the provision of more detailed examples on the importance of counter-ions in peptide activity, for example in lines 115-116, the authors could develop at least one example of this issue for any of the peptides mentioned.
- In line 206, examples of hydrophobic counter-ions should be provided.
- In general, for all the delivery systems described, it would be advisable to add information on examples of real uses with their corresponding references.
- Whenever possible, add information about the physical chemistry behind the phenomenon described, even if only superficially. This is to prevent the manuscript from being merely a phenomenological description of the effect of counter-ons on peptides.
Author Response
Reviewer 2
The manuscript from Ying Liu et al. is a review dealing with the role that counter-ions have on the structure, activity and applications of peptides. This information is key to optimizing their use as pharmacological drugs.
The manuscript is interesting and well-structured, covering the types of counter-ions and their effects on the structure, stability, and toxicity of peptides, as well as their impact on the systems used for peptide delivery or the techniques employed for their identification or separation. However, there are some issues that the author should address before its consideration for publication.
Although the review is very informative, it is generally too descriptive, so it lacks depth in explaining why and how certain counter-ions work, and also fails to provide more detailed examples of their uses.
Response: Thank you for your positive feedback on the structure and scope of our review, and for your valuable suggestions for its improvement. We appreciate your insightful comments regarding the need for greater mechanistic depth and more detailed examples. We have carefully considered this advice and have undertaken a comprehensive revision of the manuscript to provide more thorough explanations of the underlying principles and to incorporate specific, illustrative case studies. We believe these revisions have significantly enhanced the scientific rigor and analytical depth of the work, and we are grateful for your guidance in strengthening the manuscript. Our detailed responses to your specific points are provided below.
Comment 1: The figure captions are very short, so it is necessary to provide more information about what is described in each one. For example, in figure 2, add the names of the counter-ions shown. In figure 4, give the meaning of HIP and add the information that counter-ions are hydrophobic molecules. In figure 6, add the meaning of W/O and O/W and more details about the nanoemulsion system shown in the figure. The same for figure 7.
Response 1: Thank you for the suggestion to enhance the figure captions. we have thoroughly revised all figure captions to provide comprehensive descriptions. The caption of figure 2 at line 114-118 has been improved to “Types of counter-ions in acid radical ion form. Inorganic anion counter-ions: Cl⁻: chloride, SO₄²⁻: sulfate and PO₄³⁻: phosphate. Organic anion counter-ions: C₂H₃O₂⁻: acetate, C₃H₅O₃⁻: lactate, C₆H₁₁O₇⁻: gluconate, C₄H₂O₄²⁻: succinate, C₄H₄O₅²⁻: tartrate and C₇H₅O₃⁻: salicylate. Inorganic cationic counter-ions: Na⁺: sodium ion, Mg²⁺: magnesium ion, K⁺: potassium ion, Ca²⁺: calcium ion and NH₄⁺: ammonium ion. Organic cationic counter-ion: C₄H₁₂NO₃⁺: choline ion”.
The caption of figure 4 at line 190-195 has been improved to “Schematic diagram of the HIP. HIP is a technique where hydrophilic molecules form complexes with hydrophobic counter-ions through electrostatic interaction. A hydrophilic peptide (blue) interacts with a hydrophobic counter-ion through electrostatic attraction (dashed line). This process effectively masks the surface charge of the parent molecule, significantly increasing its overall lipophilicity, thereby facilitating encapsulation into lipid nanocarriers and enhancing membrane permeability and oral bioavailability”.
The caption of figure 5 at line 219-221 has been improved to “Schematic diagram of the peptide toxicity in TFA salt form. TFA has been shown to contribute to glioma cell proliferation, osteoblast and articular chondrocyte damage as well as acute liver injury”.
The caption of figure 6 at line 275-280 has been improved to “Schematic diagram of nanoemulsion delivery systems. W/O denotes a water-in-oil emulsion where aqueous droplets are dispersed within a continuous oil phase, while O/W represents an oil-in-water emulsion where oil droplets are dispersed in a continuous aqueous phase. These systems are composed of safe, biocompatible compounds including oil phases, aqueous phases, and surfactants. The incorporation of counter-ions can enhance stability through electrostatic and spatial repulsion effects”.
The caption of new figure 7 at line 318-322 is “Schematic illustration of SEDDs enhanced by HIP. The SEDDs preconcentrate consists of an oil phase, a surfactant, a co-surfactant, and peptide drugs that have been hydrophobically modified via HIP complexation with counter-ions. Upon aqueous dilution and gentle agitation, the pre-concentrate rapidly forms a fine nanoemulsion, which encapsulates the HIP-complexed peptides, thereby protecting them from enzymatic degradation and enhancing their oral absorption”.
The caption of figure 8 at 369-371 has been improved to “Schematic diagram of SLNs system and NLCs. SLNs comprise a solid lipid core stabilized by surfactants, providing a rigid matrix for sustained drug release. NLCs feature a blended solid-liquid lipid matrix that creates structural imperfections, enabling higher drug loading capacity”.
Comment 2: It should be convenient to add a figure describing the other delivery systems mentioned (slns, seedds, nlcs).
Response 2: Thank you for this valuable suggestion to include schematic diagrams of the discussed delivery systems. we have now added a new figure 7 at line 317 that specifically illustrates the composition and self-emulsification process of the SEDDs. SLNs and SLNs have already been described in figure 8 at line 370. Combined together, they provide a complete set of visual summaries for the key lipid-based delivery platforms covered in our review.
Comment 3: In section 7 there are no figures or schemes, but it would be convenient to add at least one to better understand this point.
Response 3: Thank you for this suggestion to improve the clarity of Section 7. We agree that a visual summary would be beneficial for the reader. In response to your comment, we have now integrated Table 2 into Section 7 at line 431. This table provides a concise comparative overview of the primary analytical techniques discussed in this section, including Capillary Electrophoresis, Ion Chromatography, Isotachophoresis, and coupled techniques like LC-MS. We believe this addition effectively summarizes the key information, fulfills the need for a schematic overview, and significantly enhances the accessibility and understanding of the analytical methods presented.
Comment 4: Regarding the provision of more detailed examples on the importance of counter-ions in peptide activity, for example in lines 115-116, the authors could develop at least one example of this issue for any of the peptides mentioned.
Response 4: Thank you for this valuable suggestion to strengthen our discussion with a specific example. We have added a specific example on pediocin PA-1 demonstrating how TFA⁻ alters peptide conformation while Cl⁻ provides a structurally benign alternative. The new text was at line 133-138: “Gaussier H et al. [13] demonstrated that TFA⁻ remained persistently associated with the pediocin PA-1 peptide, interfering with structural analysis and inducing subtle conformational perturbations, notably a slight increase in α-helical content. In contrast, Cl⁻ did not exhibit these associative or structure-altering effects, thereby providing a structurally benign alternative for purification”.
Comment 5: In line 206, examples of hydrophobic counter-ions should be provided. In general, for all the delivery systems described, it would be advisable to add information on examples of real uses with their corresponding references.
Response 5: Thank you for your insightful suggestion. We have added several examples to illustrate the widespread application of HIP at line 248-251: “For example, ion pairing semaglutide with ethyl lauroyl arginate (ELA) yielded hydro-phobic complexes that conferred enhanced lipophilicity and improved membrane permeability [42], and HIP of tobramycin with sodium docusate significantly increased lipophilicity and enabled stable SEDDS incorporation for oral delivery [43]”.
Besides, we also added examples in every delivery system to enhance the persuasiveness. Examples for nanoemulsion delivery systems were at line 259-264: “For example, B. D. da Silva et al. developed sub-100 nm nanoemulsions via ultrasound to significantly enhance the bioactivity of oregano essential oil, carvacrol, and thymol [44] and Fengting, Lei et al. also prepared pH-responsive sodium alginate (SA) hydro-gel-coated nanoemulsions to co-deliver CUR and EMO (CUR/EMO NE@SA) to achieve controlled drug release and specifically target macrophages of the colon [45]”.
The examples for SEDDs were at line 287-290: “It was reported that lipophilicity of insulin glargine (IG) was successfully increased via HIP with sodium octadecyl sulfate to enable incorporation into SEDDS [49] and a significant advancement arisen in enhancing the oral bioavailability of insulin IG through the innovative use of the polyglycerol/zwitterion-based SEDDS [50]”.
The examples for NLCs were at line 346-349: “It was indicated that the NLCs significantly enhanced the permeation and retention of quercetin within the skin layers [57] and charge converting nanostructured lipid carriers containing a cell-penetrating peptide could enhance cellular uptake [58]”.
Example for Liposomes was at line 378-383: “For instance, the strategic incorporation of multivalent counterions, such as Ca²⁺, induces the structural transformation of anionic liposomes into stable, solid nanocochleates, which effectively encapsulate and protect peptide-based therapeutics, thereby enhancing their stability and enabling efficient cellular delivery through membrane fusion mechanisms [63]”.
Comment 6: Whenever possible, add information about the physical chemistry behind the phenomenon described, even if only superficially. This is to prevent the manuscript from being merely a phenomenological description of the effect of counter-ions on peptides.
Response 6: Thank you for this critical suggestion to enhance the physicochemical foundation of our manuscript. We have implemented the corresponding revisions. The discussion on HIP now explicitly addresses the roles of electrostatic interactions and the hydrophobic effect, while the chromatographic analysis section details the mechanisms of dynamic charge masking and mitigation of secondary interactions. We are grateful for your guidance.
The revised text about HIP was at line 308-316: “The formation of HIP is driven by a combination of electrostatic interactions and the hydrophobic effect. The electrostatic attraction between the charged peptide and the oppositely charged hydrophobic counter-ion initiates the pair formation. Subsequently, the system’s overall free energy is lowered by minimizing the unfavorable contact between the hydrophobic moieties of the counter-ion and the aqueous environment. This process results in a net increase in the complex’s lipophilicity which directly facilitates partitioning into and encapsulation within the lipophilic matrices of nanocarriers, leading to the enhanced oral bioavailability observed in these systems [54]”.
The revised text of chromatographic analysis section details was at line 410-415: “In reversed-phase liquid chromatography (RPLC), hydrophobic counter-ions like TFA dynamically mask charged sites on the peptide through electrostatic interactions, forming ion pairs that increase their effective hydrophobicity and thereby modulating retention time and enhancing separation resolution [64,65]. This effect can eliminate peak splitting and improve peak shape by reducing undesirable secondary interactions with silanol groups on the stationary phase [36]”.
Round 2
Reviewer 1 Report
Comments and Suggestions for Authors
The authors have revised and improved their work, current version can be accepted.
Reviewer 2 Report
Comments and Suggestions for Authors
In my opinion, the manuscript has improved and is more complete with the changes made by the authors, so it is now ready for publication.